# Post-Translational Variants of Major Proteins in Amyotrophic Lateral Sclerosis Provide New Insights into the Pathophysiology of the Disease

**DOI:** 10.3390/ijms25168664

**Published:** 2024-08-08

**Authors:** Léa Bedja-Iacona, Elodie Richard, Sylviane Marouillat, Céline Brulard, Tarek Alouane, Stéphane Beltran, Christian R. Andres, Hélène Blasco, Philippe Corcia, Charlotte Veyrat-Durebex, Patrick Vourc’h

**Affiliations:** 1UMR 1253, iBraiN, Université de Tours, Inserm, 37000 Tours, France; lea.bedja--iacona@univ-tours.fr (L.B.-I.); elodie.richard@etu.univ-tours.fr (E.R.);; 2UTTIL, CHRU de Tours, 37000 Tours, France; 3Service de Neurologie, CHRU de Tours, 37000 Tours, France; 4Service de Biochimie et de Biologie Moléculaire, CHRU de Tours, 37000 Tours, France

**Keywords:** ALS, post-translational modifications, SOD1, TDP-43, FUS, TBK1

## Abstract

Post-translational modifications (PTMs) affecting proteins during or after their synthesis play a crucial role in their localization and function. The modification of these PTMs under pathophysiological conditions, i.e., their appearance, disappearance, or variation in quantity caused by a pathological environment or a mutation, corresponds to post-translational variants (PTVs). These PTVs can be directly or indirectly involved in the pathophysiology of diseases. Here, we present the PTMs and PTVs of four major amyotrophic lateral sclerosis (ALS) proteins, SOD1, TDP-43, FUS, and TBK1. These modifications involve acetylation, phosphorylation, methylation, ubiquitination, SUMOylation, and enzymatic cleavage. We list the PTM positions known to be mutated in ALS patients and discuss the roles of PTVs in the pathophysiological processes of ALS. In-depth knowledge of the PTMs and PTVs of ALS proteins is needed to better understand their role in the disease. We believe it is also crucial for developing new therapies that may be more effective in ALS.

## 1. Introduction

Amyotrophic lateral sclerosis (ALS) is a fatal neurodegenerative disease that causes the degeneration of the upper and lower motor neurons in the bulbar and spinal regions. It leads to progressive muscle weakness, paralysis, and death 3 to 5 years after the first symptoms appear. ALS has multiple causes, with environmental and genetic factors contributing to the onset of the neurodegenerative process [1]. One of the hallmarks of ALS is the presence of ubiquitin-positive protein aggregates in degenerating neurons [2]. This early observation suggested the involvement of post-translational modifications (PTMs), and in particular protein ubiquitination, in the pathophysiology of ALS [3,4,5].

In recent years, genetics have been shown to play a major role in the etiology of ALS. These discoveries have led to a better understanding of the pathophysiology of ALS and provide clues for new therapeutic approaches [6]. Hereditary forms of ALS, defined by the presence of pathogenic variants in one or more genes responsible for ALS, account for 20% of all cases and include familial cases (FALS, 15% of ALS cases) and sporadic cases (SALS, 85% of ALS cases). There are currently 30 causative genes [7]. Their involvement in the disease in terms of variant frequency is diverse, the five most frequently mutated genes being *C9orf72*, *SOD1*, *TARDBP*, *FUS*, and *TBK1* [1,8,9]. One in six cases of ALS can now be explained by a mutation in one of these five causative genes [10].

The involvement of the *C9orf72* gene in ALS has been demonstrated by repeat-primed PCR following GWAS studies that indicated a major risk factor for ALS on chromosome 9. It is the most frequently mutated gene in the disease (40% of FALS cases and 7% of SALS cases) [11]. The *C9ORF72* gene mutation is characterized by the expansion (>30 repeats) of a hexanucleotide repeat (HRE) of sequence GGGGCC in intron 1 of the gene [12,13]. The consequences of this HRE are a decrease in C9orf72 protein expression and the synthesis of toxic pre-mRNA and dipeptide protein from the intronic region containing the HRE [14,15].

The first gene to be identified as mutated in ALS, *SOD1* (12% FALS, 2% SALS), encodes for the superoxide dismutase 1 protein [16]. The discovery of pathogenic genetic variants of the *TARDBP* gene followed the observation of the protein it encodes, TDP-43, in ubiquitin-positive aggregates observed in the degenerating neurons of ALS patients [17,18,19]. Indeed, 97% of patients in whom these aggregates were detected at autopsy had positive ubiquitin–TDP-43 cytoplasmic aggregates. The mutation of the *FUS* gene in ALS, which codes for the Fused in Sarcoma protein, was performed on the basis of candidate and linkage analysis [20,21]. These two genes, *TARDBP* and *FUS*, both encode proteins involved in cellular RNA metabolism, including the regulation of RNA splicing and stability. Their mutation frequencies in ALS are similar, each explaining 4% of FALS and 1% of SALS. The *TBK1* gene encodes the TANK-binding kinase 1 protein; its mutation has been identified in ALS by exome sequencing (2% FALS, <1% SALS) [22].

The study of these four genes and their proteins SOD1, TDP-43, FUS, and TBK1 provides a better understanding of their physiological roles in the different cell types of the central nervous system and their roles in the pathophysiology of ALS, with the aim of developing new therapeutic approaches, for example, by targeting the molecular pathways involving these proteins but also by specifically targeting these causal genes or their expression at mRNA or protein level.

A better understanding of these four genes and their proteins is also crucial for characterizing the genetic variants identified in ALS patients. Genetic variants identified by Sanger sequencing, and more generally today by next-generation sequencing, can be benign or pathogenic. These variants are classified from 1 to 5 according to the international ACMG nomenclature (1, benign; 2, probably benign; 3, variant of unknown significance; 4, probably pathogenic; 5, pathogenic) [23]. Making sense of these genetic variants, i.e., being able to classify them correctly as 4 or 5, means being able to identify the cause of the disease in the patient and offer genetic counseling to family members. It also means providing access to rare treatments or to ongoing therapeutic trials. For example, the identification of a pathogenic *SOD1* variant could lead to the intrathecal administration of anti-SOD1 antisense oligonucleotides (ASOs), Tofersen [24,25]. A better understanding of *TARDBP* and *TBK1* could also lead to a better understanding of how they may be responsible for two pathologies, ALS and fronto-temporal dementia (FTD). Both diseases can run in the same family or even in the same patient. Around 15% of patients with ALS also have the criteria for FTD, with these patients presenting ALS-FTD [26].

A better understanding of these genes means a better understanding of the regulation of their expression, the functions of the proteins they encode, and their modifications under physiological and pathophysiological conditions. Among these modifications, post-translational modifications (PTMs) play a major role. Studies indicate that specific PTMs can have a major impact on the pathogenesis of genetic diseases [27]. By profoundly influencing protein structure and dynamics, PTMs play a crucial role in various biological processes such as localization, function, interaction with other proteins, and degradation of the targeted protein. PTMs can occur at any time in the life of a protein, just after translation, after localization in a specific area of the cell, or even before protein degradation. The disruption of PTMs or the formation of new PTMs can cause or contribute to a wide range of diseases, including neurodegenerative diseases [28]. Numerous studies have shown that PTM deregulation plays a central role in proteinopathies, with a strong involvement in protein aggregation [29,30].

The aim of this review is to explore the current knowledge of the post-translational modifications of four major proteins implicated in the pathophysiology of ALS. Knowledge of these PTMs on proteins under physiological and pathophysiological conditions will enable us to better understand the disease and its development and to devise new therapeutic avenues directly or indirectly targeting these modifications.

## 2. Post-Translational Variants (PTVs)

Once synthesized by the cellular translation machinery, proteins undergo various biochemical modifications, including PTMs, which enable them to acquire their cellular or extracellular localization and function [28,31]. These same proteins, under pathophysiological conditions and at different stages of the disease, may undergo other modifications such as PTMs altering their cellular localization, biochemical properties (such as their ability to aggregate), functions, and half-lives. These PTMs which appear or disappear under pathophysiological conditions can be referred as post-translational variants (PTVs), similar to the genetic variants, pathogenic, or risk factors, identified on genomes under pathophysiological conditions and implicated in a disease. PTVs can be the consequence of genetic variants leading to a change in an amino acid that makes PTM sites appear or disappear or in the post-translational action of numerous enzymes during a pathological process or in a pathological environment (Figure 1). These enzymes may act as a result of a nearby genetic variant, creating, for example, a new PTM consensus site or a conformation favoring their access to PTM sites. These genetic variants may be discovered in patients during molecular diagnostics or generated by site-directed mutagenesis for functional studies.

There are more than 200 different possible PTMs in eukaryotes, reversible or irreversible [30,32]. Among the most frequent PTMs are the addition or removal of small chemical groups (acetylation, phosphorylation, and methylation, for example), of small proteins (ubiquitination, SUMOylation, for example), and proteolysis (Table 1). Many of these modifications have been directly implicated in the pathophysiology of diseases, including neurodegenerative diseases.

Acetylation is a reversible chemical modification mostly regulated by lysine acyl transferases (KATs) and lysine deacetylases (KDACs) [33]. It plays a crucial role in a variety of biological processes, including the maintenance of chromatin stability as histones can be acetylated but also protein–protein interactions, the regulation of the cell cycle, cellular metabolism, and nuclear transport [34]. Acetyl, supplied by acetyl-coenzyme A, binds to the N-terminal amino acid of proteins or to lysines in proteins (Table 1). The N-terminal acetylation and lysine acetylation of proteins are considered reversible. Several families of lysine deacetylase enzymes (KDACs) remove the acetylation carried out by lysine acetylases (KATs).

Phosphorylation is a reversible chemical modification of proteins. This PTM is finely regulated by phosphorylation enzymes or kinases. It is involved in a wide range of cellular mechanisms, such as the regulation of enzyme activity and the transduction of numerous intracellular signals by modifying protein–protein interactions. The phosphorylation of a target protein by a kinase involves the transfer of a phosphate group carried by an adenosine triphosphate molecule to a receptor amino acid residue (Table 1). The main residues are serine, threonine, tyrosine, and histidine. Proline, arginine, aspartic acid, and cysteine residues can also be phosphorylated. Protein phosphorylation is reversible through the action of phosphatase enzymes [28].

Methylation is another post-translational chemical modification of proteins. It involves the addition of a small methyl group (CH3) to specific amino acids, mainly arginine and lysine residues (Table 1). The methyl group is added by one of the nine mammalian enzymes, protein arginine methyltransferase (PRMT). Lysine methylation plays a key role in epigenetic regulation and gene expression. Histones, for example, can be mono-, bi-, or tri-methylated. It can also affect other proteins with diverse roles, including DNA repair and signal transduction. Arginine methylation, on the other hand, is mainly involved in signal transduction and RNA metabolism. Protein methylation can be reversed by demethylases.

Ubiquitination is a PTM attaching to protein one or several ubiquitins, 76-residue polypeptides highly conserved in eukaryotes [35]. The attachment of ubiquitin molecules to target proteins is under the control of the enzymes of the ubiquitin pathway, which involves three steps catalyzed by nearly a thousand enzymes, contributing to its complexity. These enzymes are divided into three groups: ubiquitin-activating (E1), ubiquitin-conjugating (E2), and ubiquitin ligase (E3) enzymes. Thirty-seven genes encoding E2 ligases (17 families) are present in the human genomes [36]. The E3 ligases are the most numerous and are divided into two large groups, the Really Interesting New Gene (RING) domain-containing E3 ligases and the homology to the E6-associated protein carboxyl terminus (HECT) domain-containing E3 ligases [37]. Ubiquitination can occur on various amino acid residues on targeted proteins, but lysines are the most frequently targeted on proteins (Table 1). The type of ubiquitin binding to a protein and the number of ubiquitins bound (mono-, poly-, and multi-ubiquitination) will determine the fate of the protein, i.e., whether it is degraded by the proteasome or bound to another protein for functional purposes. The ubiquitin pathway is thus involved in many cellular mechanisms such as protein degradation by the ubiquitin proteasome system (Lysin 48-linked polyubiquitination) and apoptosis, proliferation, or intracellular trafficking (Lysin 63-linked polyubiquitination), for example. Like previous PTMs, protein ubiquitination is reversible, thanks to deubiquitinating enzymes (DUBs). Ubiquitination plays an important role in the regulation of cellular protein quality control and homeostasis, processes involved in the pathophysiology of ALS [38,39]. Motor neurons, which degenerate in ALS, seem particularly vulnerable to proteostasis dysfunction [40].

SUMOylation consists in the addition of a Small Ubiquitin-Related Modifier (SUMO) protein to a lysine residue on targeted proteins (Table 1). The SUMO protein family is formed of five paralogs in mammals, SUMO1 to 5. Similar to the ubiquitin pathway, the SUMOylation pathway is constituted by the action of three types of enzymes in succession, an E1 enzyme, an E2 enzyme (Ubc9), and one of the E3 enzymes. SUMOylation is also a reversible process, by SUMO-specific proteases (SENPs). It has many roles, involved in chromatin compaction (histone binding), nucleic acid repair, transcription regulation, protein localization in cells, and signal transduction, for example [41,42,43]. We have shown that the main proteins encoded by causal genes in ALS contain one or more potential SUMOylation sites and that the cellular SUMOylation process exerts a direct role in several pathogenic mechanisms involved in ALS, defects in protein homeostasis, glutamate excitotoxicity, hypoxia, and oxidative stress [44].

Proteolysis is a mechanism that irreversibly affects proteins. It is carried out by one of the largest families of proteins in cells, the proteases [45]. More than 500 different proteases can be encoded by the human genome. They are separated into five classes: serine, threonine, acid aspartic, cysteine, and metallo-proteases (Table 1). The cleavage of target proteins by these enzymes releases new N-term or C-term proteins or intermediate regions if multiple cleavages take place in the same protein [46]. These new proteins or peptides may cause the parent protein to lose its function, or activate it, or contain a new function of their own. This new function may be beneficial or toxic to the cell. These proteolysis mechanisms enable, for example, many inactive proteins to acquire their function or other proteins to be addressed in the right cellular compartment. There are therefore many mechanisms involving protein proteolysis, in both physiological and pathophysiological conditions. Work on APP cleavage in the pathophysiology of Alzheimer’s disease is one of the best known examples [47].

Several publications have reported PTM sites on the four major ALS proteins SOD1, TDP-43, FUS, and TKB1. As shown in Figure 2, modifications are present in a large number of amino acid residues on these four proteins under physiological (PTMs) or pathophysiological conditions (PTVs). Among these PTVs, several could be involved in the pathophysiology of ALS.

## 3. SOD1 Protein

The superoxide dismutase 1 protein (SOD1) is a protein of 154 amino acid residues expressed from the *SOD1* gene localized on chromosome 21q22.11. SOD1 is abundant in cells; it represents between 1 and 2% of total proteins [48]. It is mainly localized in the cytoplasm but is also found in the nucleus. SOD1 is a globular protein, which mainly contains a binding domain with another SOD1 protein, forming a homodimer, and a metal-binding loop (Figure 2). The SOD1 enzyme converts superoxide into hydrogen peroxide and oxygen. Its main role is thus to protect the cell against reactive oxygen species produced during respiration. It also acts as an RNA-binding protein or as an activator of the transcription factor following exposure to oxidative stress [49,50]. The PTMs of SOD1 regulate its 3D structure, its ability to bind metals, its distribution in the cell, and its enzymatic function [33].

The *SOD1* gene was the first gene involved in the pathophysiology of ALS [8,16]. Over 200 genetic variants have been identified along the entire length of the coding sequence but also in the introns [51,52,53]. Most variants in SOD1 are dominant gain-of-function mutations, affecting its post-translational processing and folding and leading to protein aggregation, which is toxic for motor neurons [54,55,56]. Indeed, neuronal aggregates positive for SOD1 protein are observed in ALS patients with mutations in the *SOD1* gene. Numerous *SOD1* gene variants identified in patients have been used to build in vitro and in vivo models to better understand the pathophysiology of the disease. The analysis of post-mortem spinal cord tissues from ALS patients shows many PTM changes [54], what we can call PTVs, with in particular variations in the levels of 15 PTMs on 14 specific SOD1 residues (9%) across 69% of ALS patients when compared to a control group. We have reported in Table 2 and Figure 2 all the amino acids’ sites of acetylation, phosphorylation, methylation, ubiquitination, and SUMOylation. We have also indicated whether these amino acids were known in the ClinVar database to be of uncertain, probably pathogenic, and pathogenic significance in ALS. SOD1 contains PTM sites all along the protein. Given that mutated amino acids in SOD1 (missense variants) are distributed along the entire length of the protein, it is interesting to note that very few of the PTM sites are mutated positions in ALS. This observation supports the idea that PTMs on these amino acids could participate in the gain of function provided by SOD1 in the disease.

### 3.1. Acetylation

The acetylation of wild-type SOD1 protein regulates its enzymatic activity by altering the surface charge [33]. The acetylation of lysine 71 is hypothesized to inactivate SOD1 by impairing its ability to connect to the superoxide dismutase copper chaperone, thereby preventing the formation of SOD1 homodimers [57]. In addition, the acetylation of lysine 123 inhibits SOD1’s ability to disrupt mitochondrial metabolism at respiratory complex I [58]. The mechanism by which SOD1 inhibits respiration is not yet clear in mammalian cells but has been shown to occur through the interaction of SOD1 with casein kinase 1-gamma in yeast. In this case, the acetylation of lysine 123 prevents the interaction between the two partners [59]. The acetylation of lysine 4 has been found to be reduced in some ALS patients [54]. This reduction has been linked to SOD1 aggregation and intercellular propagation in vitro [60]. Interestingly, this position is mutated in some patients. Patients carrying the Lys4Glu (K4E) variant exhibit limbic-onset ALS with slow progression [61].

### 3.2. Phosphorylation

SOD1 phosphorylation sites are associated with its nuclear localization, which is crucial for its role in transcription, enzymatic activity, and stability [33,62]. The phosphorylation of threonine 3 and 59 and serine 60 appears to be phosphorylation sites regulating SOD1 enzymatic activity [30]. As threonine 3 is distant from the dimer interface, Wilcox and colleagues proposed that it is not directly involved in the SOD1 dimer interface but may have consequences for protein dynamics and structure and thus impact on SOD1 stability and activity. This impact could come from an interaction with other proteins or partners [63]. Later, Fay and colleagues showed that threonine 3 phosphorylation is involved in dimer stabilization between SOD1 WT and the pathogenic variant SOD1 Ala5Val. However, how threonine 3 phosphorylation can lead to SOD1 stabilization is still unknown [64]. In addition, serine phosphorylation may also regulate SOD1 localization. Specifically, the phosphorylation of serine 60 and 99 enables SOD1 to localize in the nucleus during oxidative stress. This localization, in turn, facilitates its activity as a transcription factor, enabling SOD1 to activate the transcription of several genes encoding antioxidant proteins. The functional importance of these two phosphorylated serines was elucidated by Tsang and colleagues in yeast when they mutated these two serine residues, resulting in the inability of Sod1 to localize to the nucleus. In yeast, the Chk2-related kinase Dun1 binds Sod1 through the ROS-mediated activation of ATM/Mec1, but in humans, the protein involved in these mechanisms is still unknown [50]. A study employing a machine-learning approach identified serine 99 as the most likely SOD1 post-translational modification (PTM) with biological significance among all known SOD1 PTMs. Unmodified serine 99 could play a role in maintaining SOD1 structure, and the phosphorylation of this amino acid could potentially induce alterations in SOD1 conformation or activity [33]. Serine 99 phosphorylation has been found to be reduced in some cases of ALS [54]. Probably due to the importance of serine 99, the pathogenic Ser99Ala (S99A) variant of SOD1 observed in ALS patients is unable to relocalize to the nucleus and thus activate antioxidant genes during oxidative stress [50]. Other serine residues appear to be important. A homozygous Ser69Pro (S69P) variant was identified in a patient with a severe early-onset form of ALS. This condition is associated with a decrease in SOD1 enzymatic activity [65]. A pathogenic variant, Ser135Asn (S135N), which also suppresses this phosphorylation site, was discovered in ALS patients. It is located in the electrostatic loop in SOD1 and impaired metal binding [66]. This variant is associated with a form of ALS presenting a rapid progression and manifestations preferentially affecting lower motor neurons [67]. Fibroblasts from patients carrying this variant showed a metabolic shift toward glycolysis in vitro [68].

### 3.3. Methylation

The methylation of arginines 80 and 144 was observed by Larsen and colleagues (2016). This PTM may influence the enzymatic activity of SOD1 and its interactions with other proteins. However, it is important to note that the precise functional consequences of SOD1 methylation are still poorly understood, and the effects may vary depending on the methylation site and context [69].

### 3.4. Ubiquitination and SUMOylation

Lysine 137 is a target for ubiquitination by various ubiquitin ligases, such as Dorfin, NEDL1, or MITOL [33]. Recognition and ubiquitination occur directly for Dorfin and MITOL or via a complex with TRAPδ for NEDL1 [62]. A significant 2.5-fold increase in the ubiquitination of mutant SOD1 at lysine 92 was observed [54]. This shows that the ubiquitin–proteasome system attempts to degrade mutant SOD1 [54]. Lysine residues 10 and 76 are SUMOylation targets. The SUMOylation of lysine 76 by SUMO-1 stabilizes and stimulates the aggregation of the SOD1 variant [70]. SUMO-2/3 can also bind to lysine 76 of the SOD1 mutant, with the same pathological consequences as SUMO-1. We have shown that preventing the SUMOylation of mutant SOD1 reduces aggregate formation in vitro [71].

## 4. TDP-43 Protein

The *TARDBP* gene is localized on chromosome 1p36.22. It encodes the transactive response DNA-binding protein 43, TDP-43 (43 kDa), a ribonucleoprotein (hnRNP) of 414 amino acid residues. It contains an N-terminal domain, two RNA-binding domains (RRM1, RRM2), and a glycine-rich C-terminal domain [72] (Figure 2). Under physiological conditions, the majority of TDP-43 is localized in the nucleus, with a small fraction constantly shuttling between the nucleus and cytoplasm. In the nucleus, TDP-43 has a variety of functions, including gene expression, pre-mRNA splicing, and the autoregulation of its own mRNA. TDP-43 is also involved in the regulation of miRNA biogenesis. In response to stress conditions, when localized in the cytoplasm, TDP-43 assumes the role of controlling mRNA stability, translation, and nucleocytoplasmic transport by forming cytoplasmic ribonucleoprotein complexes known as stress granules [73].

TDP-43 aggregates are found in approximately 97% of ALS patients [74,75]. Pathogenic variants of the *TARDBP* gene have been found in 3% of ALS patients. The majority are missense variants located in the region coding the C-terminal region of TDP-43. These variants and our knowledge of the roles of TDP-43 in ALS pathophysiology support the idea that TDP-43 acts through both loss-of-function mechanisms and gain-of-function mechanisms by promoting its aggregation [56,76]. TDP-43 is subject to numerous PTVs, associated with positive and negative effects in the pathophysiology of ALS (Table 3). Studies support their involvement in the pathophysiology of the disease [77,78].

### 4.1. Acetylation

While the TDP-43 protein has a total of 20 lysine residues, their participation in the acetylation process is not well understood. It appears that lysine 145 and 192 are the two primary lysine residues subject to acetylation [29]. Acetylation seems to promote aggregation and to have negative consequences on TDP-43 function. Indeed, acetylation at lysine 145 and 192, localized in the RRM domain, decreases the RNA-binding ability of TDP-43 and leads to its pathogenic aggregation in ALS [79]. Mutants Gln331Lys (Q331K) and Asn345Lys (N345K) could have consequences on TDP-43 acetylation and so aggregation and activity as they both are predicted as possible sites of acetylation [80]. Interestingly, the deacetylation of TDP-43 at lysine positions 154 and 192 hyperphosphorylates TDP-43, promoting its aggregation [81].

### 4.2. Phosphorylation

The phosphorylation of threonine, serine, and tyrosine residues occurs throughout the protein TDP-43. The most important residues are located in the C-terminal domain and phosphorylated by casein kinase 1 or 2 [82]. The consequences of the increased phosphorylation of TDP-43 described in ALS are the subject of debate. Some studies support the idea that increased phosphorylation may increase the propensity to form aggregates, one argument being that phosphorylated TDP-43 is associated with the disruption of its normal functions and homeostasis. Conversely, other studies propose that phosphorylation may serve as an indication of a protective cellular defense mechanism, only becoming apparent once aggregation has already occurred [83]. Interestingly, the C-terminal glycine-rich domain (positions 274 to 414), which is a hot spot for mutations in TDP-43 in ALS, contains almost exclusively phosphorylation sites as PTM sites (Figure 2). It is also interesting to note that of the twenty serine phosphorylation sites in this domain, only two serines (Ser375 and Ser377) are mutated in ALS patients. The Ser375Gly (S375G) variant was identified in ALS patients associated with early-onset ALS [80]. The fact that the majority of serine phosphorylation sites in TDP-43 are not mutated in ALS suggests that their phosphorylation may have a role in the disease. Phosphorylation at serine 403, 404, 409, and 410 is known to be a feature of TDP-43 cytoplasmic aggregates [82]. Although most studies indicate that the C-terminal domain of TDP-43 is the principal domain involved in its aggregation, it is interesting to have a look at its N-terminal domain. Indeed, studies show that the N-terminal domain participates in the formation of reversible structures of TDP-43 polymers and that this contributes to its role in RNA splicing [84]. Interestingly, a phosphomimetic substitution of serine 48 in TDP-43 prevents these TDP-43 polymer formations, affects liquid–liquid phase separation (LLPS), and alters RNA splicing activity [85].

### 4.3. Methylation

TDP-43 methylation has not been extensively characterized in the literature, but databases have documented three methylated residues: arginine 42, 275, and 293. Nonetheless, the functions and regulatory mechanisms of these modifications remain unexplored [86].

### 4.4. Ubiquitination and SUMOylation

Ubiquitination and deubiquitination pathways play an important role in the involvement of TDP-43 in ALS pathophysiology [38]. Ubiquitinated and hyperphosphorylated TDP-43 proteins are a major constituent of the insoluble and mislocalized protein aggregates identified in ALS patients in post-mortem studies [74]. It has been observed that the overexpression or mutation of the UPS regulatory factor ubiquilin-2 in ALS leads to the mislocalization and aggregation of TDP-43 both in vitro and in vivo. Indeed, the aggregation of ubiquitinated TDP-43 in the brain of ALS patients may represent a molecular species that the UPS is unable to degrade appropriately. TDP-43 thus becomes more vulnerable to nuclear depletion and aggregation [72]. Several E3 ligases participate in the ubiquitination of TDP-43 such as Parkin [87] or its 35 and 25 kDa cleaved forms (see below the paragraph about truncation) such as the VHL/CUL2 complex or Znf179 [88,89]. Interestingly, the unique class 5 mutated PTM site (among sites of acetylation, phosphorylation, methylation, ubiquitination, SUMOylation, and truncation) in ALS or FTD reported to date is lysine 263. A pathogenic Lys263Glu variant (K263E) has been reported in FTD [90]. This variant, located in RRM2, is thought to cause the abnormal folding of TDP-43 and abortive proteasomal degradation. This would increase its ubiquitination at other sites and aggregation propensity, leading to a higher probability of cytoplasmic aggregate formation [91]. A mutagenesis study coupled with mass spectrometry analysis has also shown that positions K84 and K95 can be ubiquitinated and that this has an action on the nuclear transport of TDP-43 for K84 and on the phosphorylation of the C-terminal domain of TDP-43 for K95 [92]. The SUMOylation of TDP-43 may also have important roles. For example, preventing the SUMOylation of TDP-43 by mutating lysine 136 to arginine enabled us to show that this position is important for the localization of TDP-43 in the cell and for its ability to aggregate [93]. It has also been proposed that the SUMOylation of lysine 84 disrupts the NLS site, rendering the transport of TDP-43 from the cytoplasm to the nucleus ineffective [80].

### 4.5. Truncation

Cleaved forms of TDP-43, particularly TDP-35 and TDP-25, are observed in cytoplasmic TDP-43-positive aggregates [94]. Cleavage is carried out by caspases and calpains [95]. Cleavage at position D89-A90 leads to the formation of TDP-35 (C-term region of TDP-43), which no longer possesses the N-terminal domain (NTD) and disrupts the nuclear localization signal (NLS). This fragment appears to fold correctly. Cleavages at positions D169-G170 and D174-C175 are associated with the formation of the TDP-25 product, which also lacks the NTD, the NLS, and most of RNA recognition motif 1 (RRM1). Additional cleavage can occur at positions M218-D219 and E246-D247. All these cleavages disrupt or eliminate the NLS domain in TDP-43, trapping the protein in the cytoplasm and thereby promoting protein aggregation. Furthermore, pathogenic variants identified in ALS patients, such as Ala90Val (A90V) and Asp169Gly (D169G), link these cleavage sites to pathological mutated forms of TDP-43 [80].

## 5. FUS Protein

The Fused in Sarcoma (FUS) protein, also known as Translocated in Liposarcoma (TLS), is a 526-amino-acid-residue RNA-binding protein ubiquitously expressed and encoded by the *FUS* gene located in 16p11.2. As shown in Figure 2, it contains a low-complexity region enriched in serine, tyrosine, glycine, and glutamine (SYGQ) residues, an RNA recognition motif (RRM), a zinc finger domain, and three Arg-Gly-Gly-rich (RGG) domains. It also contains an NLS domain and an NES domain. Its expression is predominantly nuclear. It is involved in DNA repair and the regulation of gene expression at various stages, transcription, splicing, and RNA transport, for example [96].

Around 60 pathogenic variants have been reported in the *FUS* gene to date in ALS, the majority located in the C-terminal region containing the NLS domain encoded by the exon 15 [97,98]. These variants, most of which act via a dominant gain-of-function mechanism, lead to the nuclear depletion of FUS, its cytoplasmic accumulation, and the formation of aggregates (Table 4). Some of the pathogenic FUS variants are responsible for the most aggressive, juvenile-onset forms of ALS [99,100].

### 5.1. Acetylation

When the CREB-binding protein (CBP/p300) acetylates the lysine 510 in FUS, located in its NLS domain, TRN1 is no longer able to import FUS into the nucleus, leaving it in the cytoplasm where it assembles into stress granule-like inclusions [101]. The CBP/p300 acetylation of lysines 315 and 316, two amino acid residues located in the RRM domain, can prevent FUS from binding to its target RNAs [101]. Arenas and colleagues also found an approximately 50% increase in FUS protein levels acetylated at lysine 510 in fibroblasts derived from ALS patients compared with fibroblasts from healthy controls. So, they hypothesized that there might be a link between FUS protein acetylation and the pathological features of FUS-related ALS [101]. This observation, coupled with the presence of pathogenic variants in ALS patients at this precise location, Lys510Gln (K510E), Lys510Arg (k510R), and Lys510Met (K510M), supports the idea that lysine 510 acetylation must play an important role in FUS function and its role in ALS [102,103,104].

### 5.2. Phosphorylation

The phosphorylation of FUS has different functions depending on the domain targeted. The phosphorylation of residues in the N-terminal region appears to affect FUS self-association, whereas C-terminal phosphorylation induces changes in FUS cellular localization [105]. Indeed, the phosphorylation of tyrosine 526 by a Src family kinase leads to its cytoplasmic accumulation. This is due to a decrease in the binding affinity between FUS and its nuclear import receptor, transportin-1 [106]. Studies have shown that this phosphorylation occurs in response to DNA damage to facilitate the cytoplasmic localization of FUS [105]. Interestingly, we previously identified a pathogenic Tyr526Cys variant (Y526C) at this position in a 25-year-old woman with rapidly progressing bulbar-onset ALS [99]. Phosphorylation can also reduce protein aggregation. Deletions of serines 57 and 96 are associated with ALS [107,108]. With the existence of these ALS-causing variants and the discovery of serine 96 phosphorylation, the idea that decreased phosphorylation capacity could increase FUS’s propensity for aggregation was proposed [109].

### 5.3. Methylation

FUS localization is also modulated by methylation. The arginine methylation of the RGG domains regulates FUS binding to transportin and thus its nuclear import [105]. Several pathogenic variants have been identified in ALS patients on arginine residues located in the C-terminal NLS domain of the FUS protein. Variants have been identified at position R514 (Arg514Gly, Arg514Ser) and R518 (Arg518Lys) [20,21,110]. Studies of the pathogenic R521C variant in hIPSCs-derived neurons showed an increase in FUS polyubiquitination and the formation of aggregates [111]. Functional studies demonstrate that Arg524Ser (R524S) identified in ALS patients disrupts the interaction between the FUS protein and its nuclear import receptor [112,113].

### 5.4. Ubiquitination and SUMOylation

The high-throughput proteomic analysis of various cell lines has revealed that the ubiquitination of FUS can take place in its RRM or ZnF domains, which could contribute to FUS degradation [114]. The ubiquitination and SUMOylation of RNA-binding proteins (RBPs) serve to prevent aggregation by maintaining RBPs in a soluble state [115]. FUS inclusions found in the cytoplasm of ALS patients exhibit high levels of ubiquitin. This disrupts the usual balance of ubiquitin in cells, potentially leading to a dysregulation of protein quality control pathways [4]. A previous study also highlighted the importance of SUMOylation in facilitating the degradation of the ALS-associated FUS mutant Pro525Leu (P525L) variant. This degradation is mediated by the ubiquitin–proteasome system (UPS), inhibiting its accumulation within stress granules. The SUMO-targeted ubiquitin ligase (StUbL) pathway may link the SUMOylation of RNA-binding proteins (RBPs) in the nucleus to SG disassembly in the cytosol. A defect in this mechanism has been shown to exacerbate pathological aggregation in this FUS mutant [116].

## 6. TBK1 Protein

The serine/threonine kinase protein TANK-binding kinase 1 (TBK1) protein is expressed from the *TBK1* gene located at 12q14.2. The TBK1 protein comprises an N-terminal kinase domain (KD), a ubiquitin-like domain (ULD), two coiled-coil domains (CCD1, CCD2), a zinc finger domain, and an HLH region (Figure 2). It plays crucial roles in the control of several physiological processes such as innate immunity, autophagy, mitochondrial metabolism, cell survival, and proliferation [117].

The involvement of the *TBK1* gene in ALS has been found by exome sequencing [22,118] (Table 5). The pathogenic variants are located all along the gene. They act primarily through loss-of-function mechanisms in autophagy and mitophagy, although gain-of-function actions may also be suggested via OPTN-related aggregation formation. Several pathogenic variants have shown an important role for TBK1 in neuroinflammatory signaling pathways [119].

### 6.1. Acetylation

TBK1 activity is regulated by lysine acetylation; indeed, the removal of acetyl groups from lysine 241 and 692 by HDAC3 plays a crucial role in enhancing the kinase activity and dimerization of TBK1 [120].

### 6.2. Phosphorylation

The trans-autophosphorylation of serine 172 induces TBK1 activation via a change in its conformation. Phosphatases then reduce TBK1 activity by removing the activating phosphorylation [121]. Tyrosine kinase Src promoted the phosphorylation of TBK1 on Tyr 179. Interestingly, the mutagenesis of this position tyrosine 179 to alanine resulted in the impaired autophosphorylation of TBK1 at Ser 172, which is required for TBK1 activation [122,123]. A reduction in TBK1 phosphorylation was observed in lymphoblasts of an ALS patient carrying a pathogenic variant p.Arg573Gly [124]. The phosphorylation of serine 527 by dual-specificity tyrosine-(Y)-phosphorylation-regulated kinase 2 (DYRK2) induces TBK1 polyubiquitination and degradation [125].

### 6.3. Methylation

Little is known about TBK1 methylation. It has recently been shown that the arginine methylation of TBK1 promotes its activation. PRMT1, protein arginine methyltransferase type I (PRMT1), regulates TBK1 activation by inducing its methylation at several positions, Arg54, Arg134, and Arg228. These PTMs enhance TBK1 oligomerization after viral infection, thereby promoting TBK1 phosphorylation [126]. A recent study also showed that the lysine methyltransferase SETD4 induces the methylation of TBK1 on lysine 607 [127].

### 6.4. Ubiquitination and SUMOylation

Ubiquitination is a central PTM for the regulation of TBK1 activity. TBK1 signaling is limited by the polyubiquitination of lysine 48 by E3 ligases which leads to TBK1 degradation. A deubiquitinase complex removes lysine 48 polyubiquitination, preventing the degradation of the TBK1 protein [122]. SUMOylation on lysine 694, located near the C-terminal domain, is known to contribute to TBK1 antiviral activity [128]. Further studies will be needed to better understand the role played by these PTMs in TBK1 activity and PTVs in ALS.

## 7. Discussion and Conclusions

In ALS, PTVs may participate directly or indirectly in the pathogenesis of the disease by acting on several mechanisms, such as protein aggregation, mislocalization in the cell, and loss of enzymatic function, for example. In this review, we present the main PTMs affecting four major ALS proteins. These proteins are encoded by genes mutated in familial and sporadic forms of the disease. In addition to the documented PTMs and PTVs presented in this review, there could be unidentified PTVs associated with specific genetic variants. Indeed, alterations in the amino acid sequence from these four proteins have the potential to introduce new PTV sites, either directly through changes in amino acids due to genetic variants in patients or indirectly via conformational shifts that expose PTM sites that are typically inaccessible. The genetic variants in the four proteins SOD1, TDP-43, FUS, and TBK1 may have consequences on their structure and functions, as mentioned previously in this review.

The first observation to notice is the large number of PTMs localized along these four proteins SOD1, TDP-43, FUS, and TBK1 and therefore the considerable complexity that must exist in the roles played by these PTMs (physiologic conditions) or PTVs (physiopathological conditions) on protein folding, localization, interactions, and functions. It is well known that PTMs, and probably PTVs too, act even when small fractions of proteins are modified in cells. This of course complicates their studies under physiological or pathophysiological conditions. The fact that the same position in a protein, a lysine, for example, can be modified by different types of PTMs does not make things any easier either. The fact that these PTMs, with the exception of cleavage, are reversible also adds complexity to the studies. Moreover, this review is not exhaustive of the diversity of PTMs that exist. There are over 200 different types of PTMs in eukaryotes. The four proteins mentioned, SOD1, TDP-43, FUS, and TBK1, contain other PTMs. These include oxidation, amidation, and citrullination, in particular [30]. Variations in the quantity of some of these PTMs in ALS, i.e., PTVs, have been found in SOD1 in tissues from ALS patients compared with control individuals [54]. Concerning TDP-43, in vitro studies support the idea that all these cysteines could be targets of oxidation. Interestingly, pathogenic genetic variants identified in patients and contributing new cysteines, G348C (Gly348Cys) and S379C (Ser379Cys), have been shown to be targets of oxidation and to be associated with a greater accumulation of disulphide cross-linked species of TDP-43 [129].

Acting on the PTVs of proteins seems to be a very interesting way of acting on the pathophysiological pathways involved in ALS. For example, the administration of small chemical molecules such as acetylation inhibitors (sodium butyrate and trichostatin) to ALS model mice (SOD1 G94A mutant) is associated with neuroprotection, and the action of small molecules on the SUMOylation levels affects TDP-43 aggregation [81,130]. The development of antibodies or ScFvs targeting proteins carrying particular PTMs or PTVs is also an interesting approach to test in ALS. Post-translational intracellular silencing antibody (PISA) technology could be generated against the major ALS proteins. This approach consists of a method that allows us to select intrabodies against post-translationally modified proteins [131]. The goal with these molecules would be, for example, to enable proteins carrying PTVs to reposition correctly in the cell in order to recover all or part of their functions or to avoid forming aggregates. PTMs could serve as biomarkers in neurodegenerative diseases such as ALS or TDP-43 proteinopathies [132].

A better characterization of the PTMs of the proteins SOD1, TDP-43, FUS, and TBK1, and of the PTVs occurring on these proteins during the pathophysiological processes involved in ALS, is very important for a better understanding of the diverse roles of these four major proteins in neuronal cells and in ALS pathophysiology. It is also crucial to continue these studies on PTMs and PTVs in order to identify particular forms of these proteins that could be targeted in the objective of translational research programs, to develop effective therapeutics for this incurable neurodegenerative disease. Indeed, this review shows a large number of PTMs in the principal proteins implicated in ALS and the description of several PTVs involved in the etiology of the disease. It also shows how PTVs specific to very precise sites in proteins affect the function, half-life, or localization of a protein, as, for example, the SUMOylation of a precise position can modify the cellular localization of a protein and/or its aggregation. The enzymes responsible for these PTVs are therapeutic targets, as are the PTV sites themselves. The aim would be to increase or decrease these particular enzymatic activities by small chemical molecules or peptides, for example. Regions of proteins carrying these PTVs could be targeted by antibodies to counter their effects. We have also indicated the sites of pathogenic genetic variants causing PTVs in ALS. We believe that specific studies should be carried out on these particular positions, as this work could identify key sites for targeting these proteins in the disease.

## Figures and Tables

**Figure 1 ijms-25-08664-f001:**
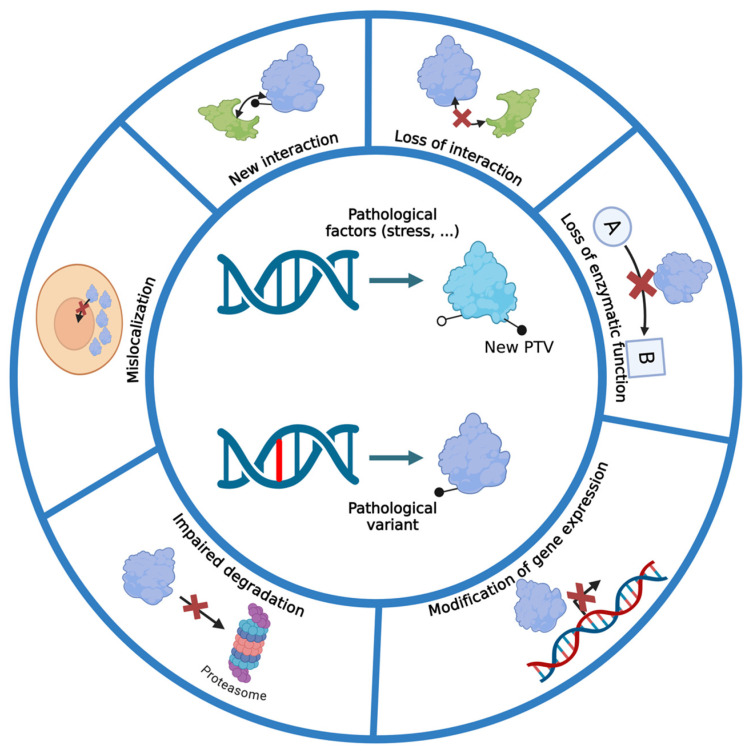
Functional consequences of PTV (chemical modification or cleavage): WT protein or mutated protein (pathological variant) with a new PTV (black dots) and consequences on protein interaction, function, localization, and half-life.

**Figure 2 ijms-25-08664-f002:**
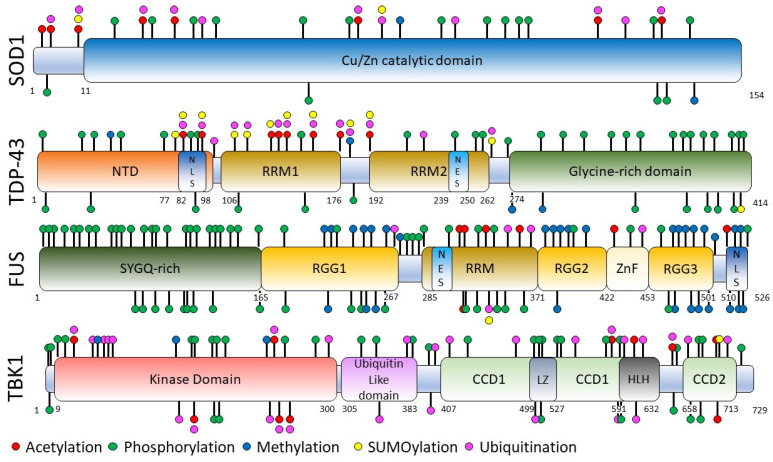
Schematic representations of SOD1, TDP-43, FUS, and TBK, with their post-translational modification sites.

**Table 1 ijms-25-08664-t001:** Post-translational modifications (PTMs), amino acids targeted in proteins, and mechanisms by which they are targeted. In bold, preferably modified amino acids.

PTMs	Targeted Amino Acids	Mechanisms
**Phosphorylation**	**Ser**, **Thr**, **Tyr**, **His**, Pro, Arg, Asp, Cys	Transfer of a phosphate from ATP by a kinase.
**Acetylation**	**Lys**, Ala, Arg, Cys, Gly, Glu, Met, Pro, Ser, Thr, Val	Transfer of an acetyl from acetyl CoA by acetyltransferase or histone acetyltransferase.
**Methylation**	**Lys**, **Arg**, Ala, Asn, Asp, Cys, Glu, Gln, His, Leu, Met, Phe, Pro	Addition of a methyl group by a methyltransferase.
**Ubiquitination**	Preferentially **Lys**	Transfer of ubiquitin by ubiquitin-conjugating (E2) enzymes or ubiquitin ligases (E3).
**SUMOylation**	**Lys**	Transfer of SUMO protein by SUMO-conjugating (E2) and SUMO ligases (E3).

**Table 2 ijms-25-08664-t002:** PTM in SOD1 protein: PTM sites based on information on dbPTM and PhosphoSitePlus websites and in papers discussed in this review. Positions with genetic variants described in ALS patients are underlined: not bold (class 3, variant of uncertain significance) and bold (class 4 or 5, variant likely pathogenic or pathogenic, respectively). Classes are according to ACMG [23].

PTM in SOD1	Amino Acid	Position in the Protein
Acetylation	Lysine	**4**, 10, 24, **31**, 71, 123, 137
	Alanine	2
Phosphorylation	Serine	26, 35, **60**, **69**, 99, 103, 106, 108, **135**, 143
	Threonine	3, 18, 40, 59, 89, 136, **138**
Methylation	Arginine	80, 144
Ubiquitination	Lysine	4, 10, 24, **31**, 37, 71, 76, 92, 123, 129, 137
SUMOylation	Lysine	10, 76

**Table 3 ijms-25-08664-t003:** PTM in TDP-43 protein: PTM sites based on information on dbPTM and PhosphoSitePlus websites and in papers discussed in this review. Positions with genetic variants described in ALS patients are underlined: not bold (class 3, variant of uncertain significance) and bold (class 4 or 5, variant likely pathogenic or pathogenic, respectively). Classes are according to ACMG [23].

PTM in TDP-43	Amino Acid	Position in the Protein
Acetylation	Lysine	84, 95, 136, 140, 145, 154, 160, 176, 192
Phosphorylation	Threonine	25, 30, **32**, **88**, 116, 153
	Tyrosine	4, 73, 155, 214
	Serine	2, 48, 91, 92, 183, 242, 254, 258, 273, 292, 305, 317, 333, 342, 347, 350, 369, **375**, **377**, 379, 387, 389, **393**, 395, 403, 404, 407, 409, 410
Methylation	Lysine	79, 84, 95, 114, 121, 136, 145, 160, 181, 192, **263**, 408
Ubiquitination	Lysine	84, 95, 102, 114, 121, 140, 145, 160, 176, 181, 192, 224, **263**
	Lysine	181
	Arginine	42, 275, 293
SUMOylation	Lysine	84, 95, 136, 140, 145, 160, 176, 192
Truncation	Threonine	25, 30, **32**, **88**, 116, 153

**Table 4 ijms-25-08664-t004:** PTM in FUS protein: PTM sites based on information on dbPTM and PhosphoSitePlus websites and in papers discussed in this review. Positions with genetic variants described in ALS patients are underlined: not bold (class 3, variant of uncertain significance) and bold (class 4 or 5, variant likely pathogenic or pathogenic, respectively). Classes are according to ACMG [23].

PTM in FUS	Amino Acid	Position in the Protein
Acetylation	Lysine	312, 315, 316, 332, 357, 427, **510**
Phosphorylation	Serine	3, 26, 30, 37, 42, 54, 57, 61, 77, 84, 86, 87, 95, 96, 108, 110, 112, 115, 117, 127, 129, 131, 135, 142, 148, 163, 164, 182, 183, 221, 257, 273, 277, 282, 340, 346, 360, 439, 462
	Tyrosine	232, 239, 304, 325, 397, 468, **526**
	Threonine	7, 11, 19, 68, 71, 78, 109, 286, 317, 326, 338
Methylation	Arginine	213, **216**, 218, 234, 242, 244, 248, 251, 259, 269, 377, 383, 386, 388, 394, 407, 472, 473, 476, 481, 485, 487, 491, 495, 498, 503, **514**, **518**, **521**, 522, **524**
	Lysine	365
Ubiquitination	Lysine	264, 316, 334, 348, 357, 365, 448
SUMOylation	Lysine	334, 357

**Table 5 ijms-25-08664-t005:** PTM in TBK1 protein: PTM sites based on information on dbPTM and PhosphoSitePlus websites and in papers discussed in this review. Positions with genetic variants described in ALS patients are underlined: bold (class 4 or 5, variant likely pathogenic or pathogenic). Classes are according to ACMG [23].

PTM in TBK1	Amino Acid	Position in the Protein
Acetylation	Lysine	30, 154, 236, 241, 251, 584, 607, 646, 691, 692
Phosphorylation	Serine	3, 5, 12, **151**, 172, 247, 509, 510, 511, 527, 531, 716
	Threonine	4, 20, 176, 278, 503, **664**, 672, 674
	Tyrosine	153, 174, 179, 185, 325, 340, 354, 394, 435, 577, 591, 592, 647, 650, 677
Methylation	Arginine	54, 134, 228
	Lysine	607
Ubiquitination	Lysine	30, 48, 60, 65, 69, 137, 154, 231, 236, 241, 251, 291, 344, 372, 396, 401, 416, 484, 504, 545, 584, 589, 596, 608, 615, 646, 661, 702
SUMOylation	Lysine	694

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
