# Peer review of "Post-Translational Variants of Major Proteins in Amyotrophic Lateral Sclerosis Provide New Insights into the Pathophysiology of the Disease"

_ijms, 2024, doi:10.3390/ijms25168664_

Round 1

Reviewer 1 Report

Comments and Suggestions for Authors

The article needs more coherence in its structure. There are repetitive sections, such as discussions on the importance of PTMs, which are scattered throughout the article and can be presented in a single dedicated section. Although the overall scope of the article is broad, a deeper analysis in certain areas could improve it, such as developing specific post-translational variants (PTVs) that influence the pathophysiology of amyotrophic lateral sclerosis (ALS).

The tables and figures need to be clearer and more explanatory because they are difficult to interpret. The captions should be more detailed with a simpler presentation.

The literature needs to be improved as some cases lack recent and relevant references. For example, in the discussion about SOD1 modifications, the last cited reference is from 2013, which does not accurately reflect the most recent developments in this area. More current studies on the modifications and their impact on SOD1 function should be included.

Additionally, in the section on TBK1, although some studies from 2020 and 2022 are mentioned, the discussion could be improved with more recent research from 2023 that provides new perspectives on the functional and structural consequences of TBK1 variants in ALS and frontotemporal lobar degeneration.

Similarly, the discussion on TDP-43 aggregation and toxicity includes references up to 2018 but could be updated with recent research analyzing how post-translational modifications influence these properties. An update of references is also needed in the general review of post-translational modifications (PTMs), as most references date before 2020.

The conclusions need to be more decisive and clear about the practical implications of the findings. They should briefly summarize the most important results of the study, highlighting how specific post-translational modifications influence the pathophysiology of ALS.

The possible clinical applications of these findings should also be discussed. How could this research develop new targeted therapies? Specific recommendations for clinical practice based on the study's findings could also be made, such as considering post-translational modifications as potential biomarkers for early diagnosis and monitoring ALS progression.

It is essential to emphasize the promotion of translational research that uses these findings to develop and test new therapies aimed at correcting pathogenic post-translational modifications in clinical trials.

Reviewer 2 Report

Comments and Suggestions for Authors

The authors present an insightful analysis of post-translational modifications (PTMs) and post-translational variants (PTVs) on key proteins associated with amyotrophic lateral sclerosis (ALS). They briefly introduce the clinical features of ALS, the key role of genetics in its etiology, and the importance of post-translational variants in the pathophysiology of the disease. The authors review the PTMs and PTVs of 4 major proteins implicated in the pathophysiology of ALS, namely SOD1, TDP-43, FUS, and TBK1. These modifications include acetylation, phosphorylation, methylation, ubiquitination, SUMOylation and enzymatic cleavage. They list all modifications for each protein in precise tables. The review focuses on PTMs and PTVs in ALS with detailed information and will therefore be of great benefit to the readers of this journal. The paper is clear and easy to read, figures and tables are easy to understand. The references are appropriate. erences are adequate.
